# Whole Locus Sequencing Identifies a Prevalent Founder Deep Intronic *RPGRIP1* Pathologic Variant in the French Leber Congenital Amaurosis Cohort

**DOI:** 10.3390/genes12020287

**Published:** 2021-02-18

**Authors:** Isabelle Perrault, Sylvain Hanein, Xavier Gérard, Nelson Mounguengue, Ryme Bouyakoub, Mohammed Zarhrate, Cécile Fourrage, Fabienne Jabot-Hanin, Béatrice Bocquet, Isabelle Meunier, Xavier Zanlonghi, Josseline Kaplan, Jean-Michel Rozet

**Affiliations:** 1Laboratory of Genetics in Ophthalmology (LGO), INSERM UMR1163, Institute of Genetics Diseases, Imagine and Paris Descartes University, 75015 Paris, France; xaviergerard26@yahoo.fr (X.G.); nelson.mounguengue@gmail.com (N.M.); ryme.bouyakoub@orange.fr (R.B.); josseline.kaplan@inserm.fr (J.K.); jean-michel.rozet@inserm.fr (J.-M.R.); 2Translational Genetics, Institute of Genetic Diseases, INSERM UMR1163, Imagine and Paris Descartes University, 75015 Paris, France; sylvainhanein@hotmail.com; 3Genomics Platform, Institute of Genetics Diseases, Imagine and Paris Descartes University, 75015 Paris, France; mohammed.zarhrate@institutimagine.org; 4Bioinformatic Platform, Institute of Genetic Diseases, Imagine and Paris Descartes University, 75015 Paris, France; cecile.fourrage@institutimagine.org (C.F.); fabienne.hanin@parisdescartes.fr (F.J.-H.); 5Bioinformatics Core Facility, Université Paris Descartes-Structure Fédérative de Recherche Necker, INSERM US24/CNRS UMS3633, 75015 Paris, France; 6Centre de Référence des Affections Sensorielles Génétiques, Institut des Neurosciences de Montpellier, CHU-Saint Eloi Montpellier, 34091 Montpellier, France; beatrice.bocquet@inserm.fr (B.B.); isabelle.meunier@inserm.fr (I.M.); 7National Reference Centre for Inherited Sensory Diseases, Univ Montpellier, CHU, 34091 Montpellier, France; 8Eye Clinic Jules Verne, 44300 Nantes, France; secretaire.zanlonghi@ophtalliance.fr; 9CHU, 35000 Rennes, France; 10Ophthalmology Department, University Hospital Henri Mondor, APHP, 94000 Créteil, France

**Keywords:** Leber congenital amaurosis, *RPGRIP1*, deep intronic variant, oligotherapy

## Abstract

Leber congenital amaurosis (LCA) encompasses the earliest and most severe retinal dystrophies and can occur as a non-syndromic or a syndromic disease. Molecular diagnosis in LCA is of particular importance in clinical decision-making and patient care since it can provide ocular and extraocular prognostics and identify patients eligible to develop gene-specific therapies. Routine high-throughput molecular testing in LCA yields 70%–80% of genetic diagnosis. In this study, we aimed to investigate the non-coding regions of one non-syndromic LCA gene, *RPGRIP1*, in a series of six families displaying one single disease allele after a gene-panel screening of 722 LCA families which identified 26 biallelic *RPGRIP1* families. Using trio-based high-throughput whole locus sequencing (WLS) for second disease alleles, we identified a founder deep intronic mutation (NM_020366.3:c.1468-128T>G) in 3/6 families. We employed Sanger sequencing to search for the pathologic variant in unresolved LCA cases (106/722) and identified three additional families (two homozygous and one compound heterozygous with the NM_020366.3:c.930+77A>G deep intronic change). This makes the c.1468-128T>G the most frequent *RPGRIP1* disease allele (8/60, 13%) in our cohort. Studying patient lymphoblasts, we show that the pathologic variant creates a donor splice-site and leads to the insertion of the pseudo-exon in the mRNA, which we were able to hamper using splice-switching antisense oligonucleotides (AONs), paving the way to therapies.

## 1. Introduction

Inherited Retinal Dystrophies (IRDs) encompass a vast group of rare monogenic degenerative diseases of photoreceptor cells and are the leading cause of blindness in children and young adults in Europe [1]. Leber congenital amaurosis (LCA) gathers a group of IRDs manifesting in severe vision impairment or blindness by one year of age. The visual outcome in LCA cases is variable ranging from cone–rod dystrophy with poor visual function to rod–cone dystrophy with some transitory visual acuity. Achromatopsia (ACHM) and congenital stationary night blindness (CSNB) are two slowly progressive retinal diseases that can present in the same way at birth. ERG is a critical test for early differential diagnosis as ERG responses in LCA but not ACHM and CSNB, are undetectable in keeping with an extremely severe rod and cone dysfunction. However, ERG recordings might be challenging in young children, and this might contribute to a diagnosis ambiguity or misdiagnosis. Early diagnosis is pivotal to forecasting both the visual and extraocular outcomes, knowing that LCA can be the initial symptom in a spectrum of multisystemic ciliopathies [2] and, more rarely, some devastating neuro–metabolic disorders [3] or tubulinopathies [4]. The identification of the underlying genetic defect can accelerate an early differential diagnosis because there is no genetic overlap between a stationary and degenerative retinal disease and very little overlap between non-syndromic and syndromic LCA. The importance of genetic diagnosis in the advent of recently evolving therapies for inherited retinal degenerations is warranted [5]. However, the diagnosis yield of the sequencing of coding regions and intron–exon boundaries of known genes is limited to 50% to 80%. While yet unknown genes are likely to be identified, studying of the non-coding regions of known disease-causing genes would certainly improve the diagnosis yield and thus patient care.

*RPGRIP1* encoding the retinitis pigmentosa GTPase regulator protein (RPGR)-Interacting Protein-1 is one of the 20 LCA genes listed in the Online Mendelian Inheritance in Man (OMIM) database (PS204000). It encodes a protein which primarily localises in the transition zone of photoreceptor cells known as the connecting cilium [6]. It is also detected in the testis but not in organs that are typically involved in multisystemic ciliopathies (e.g., brain, kidney, bones, liver etc.) [7]. Since the protein is almost exclusively expressed in the retina, pathologic variants in *RPGRIP1* cause a non-syndromic form of LCA (LCA6, (MIM605446)). Gene-panel based screening for mutations in non-syndromic and syndromic LCA and differential diagnosis genes in our cohort of 722 LCA families identified 26 and 6 index cases with biallelic and monoallelic *RPGRIP1* mutations, respectively.

Here, using a combination of trio-based high-throughput whole locus sequencing in monoallelic cases and Sanger sequencing analysis of non-coding variants in unresolved families (106/722), we identified a founder deep intronic mutation (c.1468-128T>G) in six families (two homozygous; four compound heterozygous including one with another deep intronic change c.930+77A>G; 8/60 *RPGRIP1* disease alleles), which predicts the absence of systemic involvement. 

In our study, we made an attempt at antisense oligonucleotide (AON) treatment of the lymphoblasts obtained from patients carrying the change. We were able to correct the abnormal splicing, which could have an important insight into splice-switching therapy of this frequent *RPGRIP1* disease allele (13%).

## 2. Materials and Methods

### 2.1. Families

Twenty-four individuals from 9 simplex families were included in the study (Figure 1). All individuals or legal representatives consented to the study, which received approval from the Institutional Review Boards *Comité de Protection des Personnes Ile de France II (Necker)*. All the patients in this study were referred to the Laboratory of Genetics in Ophthalmology at *Imagine* by an ophthalmologist of one of centers of reference for rare eye diseases of the national network SENSGENE (www.sensgene.com, accessed on 17 February 2021) for symptoms suggestive of LCA without overt extra-ocular involvement. The clinical characteristics are outlined in Table 1.

### 2.2. Next-Generation Sequencing

The target regions of 55 genes, the mutations of which cause LCA and ciliopathies with LCA as the initial symptom, ACHM and CSNB were used to identify mutations in LCA patients (Appendix A). SureSelect panel 1′ were 468 kb, covered 966 exons, 1000 bases of 5′ and 3′ UTRs and 50 bp spanning splice junctions (Agilent, Les Ullis France). 

The whole genomic sequence of *RPGRIP1* (2 kbp upstream 5’UTR, 2 kbp downstream 3′ UTR, coding regions and intronic sequences) was targeted and captured using 2000 120-bp biotinylated complementary RNA probes designed with the Agilent SureDesign web application (H. sapiens, hg19, GRCh37, February 2009). In order to improve the capture of both GC-rich and repetitive regions, the parameters of probe selection were adapted (Tiling density 5 × and boosting Maximize Performance Least Stringent Masking or No Masking, respectively). The targeted and enriched regions of interest were pulled out with magnetic streptavidin beads, amplified by PCR with indexing primers and sequenced on an Illumina HiSeq2500 HT system (paired-end sequencing, 2 × 130 bases) on the Genomic Platform at the Imagine Institute, Paris.

Data analysis was performed with Paris Descartes University/Imagine Institute’s Bioinformatics core facilities. Paired-end sequences were mapped on the human genome reference (NCBI build37/hg19 version) using the Burrows–Wheeler Aligner. Downstream processing was carried out with the Genome Analysis Toolkit (GATK), SAMtools [8], according to documented best practices (http://www.broadinstitute.org/gatk/guide/topic?name=best-practices, accessed on 17 February 2021). Variant calls were made with the GATK Unified Genotyper based on the 72nd version of ENSEMBL database. Genome variations were defined using the in-house software PolyDiag for gene-panel analysis, which filters out irrelevant and common polymorphisms based on frequencies (minor allele frequency ≥ 0.01) extracted from the Genome Aggregation Database (gnomAD; https://gnomad.broadinstitute.org/, accessed on 17 February 2021) developed by an international coalition of investigators, with the goal of aggregating and harmonising both exome and genome sequencing data from a wide variety of large-scale sequencing projects. Consequences of variants were predicted using: Polyphen2 [9], SIFT [10], Mutation Taster [11], NNSLPICE, MatEntScan and SpliceSiteFinder available through Alamut-Visual mutation analysis software (https://www.interactive-biosoftware.com/alamut-visual/, accessed on 17 February 2021) and the deep-learning-based tool to identify splice variants Splice AI (https://github.com/Illumina/SpliceAI, accessed on 17 February 2021) [12]. To evaluate copy number variations (CNVs, i.e., duplication and large deletion events for each individual) the relative read count for each targeted region was determined as the ratio of the read count for that region divided by the total absolute read counts of all targeted regions of the design. The ratio of the relative read count of a region in a given individual over the average relative read counts in other individuals of the run resulted in the estimated copy number for that region in that individual (method adapted from [13]).

Variant numbering is based on the *RPGRIP1* reference sequence NM_020366.3.

### 2.3. Cell Culture

Lymphoblasts were obtained from one LCA patient harbouring the c.1468-128T>G and c.1502_1505dup pathologic variants in compound heterozygous (LCA215) and two control individuals (C1 and C2). Lymphoblasts were proliferated at 37 °C, 5% CO_2_ in RPMI medium (Thermo Fisher Scientific, Courtaboeuf,, France) supplemented with 10% foetal bovine serum (Thermo Fisher Scientific, Courtaboeuf, France) and 1% streptomycin/penicillin (Thermo Fisher Scientific, Courtaboeuf, France).

Skin biopsies were obtained from two LCA patients harbouring the c.1468-128T>G pathologic variant in homozygosity (LCA426) and compound heterozygosity with the c.1525C>T substitution (MON035) and 3 control individuals (C1 to C3). Primary fibroblasts were isolated by selective trypsinisation and proliferated at 37 °C, 5% CO_2_ in Opti-MEM Glutamax I medium (Thermo Fisher Scientific, Courtaboeuf, France) supplemented with 10% fetal bovine serum (Thermo Fisher Scientific, Courtaboeuf, France), 1% ultroser G substitute serum (Pall, Saint-Germain-en-Laye, France), and 1% streptomycin/penicillin (Thermo Fisher Scientific, Courtaboeuf, France).

### 2.4. Design of AON and AON Treatment

Important parameters including guanine–cytosine content influence the efficiency of AONs to induce exon skipping [14]. We designed four 2′-O-methyl-phosphorothioate (2′-OMePs) AONs using the ESEfinder program to target the acceptor site (AON1: 5′-CUUUUCAAGUCCUCAUCUGA-3′), an exonic splice enhancer (ESE, AON2: 5′-AGCAGUGAAGGGAGAUGAUACA-3′) and the donor site (AON3: 5′-AACACGGACCUGUAUGACU-3′), respectively. A sense version of the AON1 was used as a control (AONsense 5′-UCAGAUGAGGACUUGAAAAG-3′). Lymphoblasts of Patient LCA215 and controls C1 and C2 were transfected with 2′-OMePS AON1, 2, 3 or AONsense (150 nmoL/L) in RPMI using Lipofectamine 2000 (Thermo Fisher Scientific, Courtaboeuf, France) at a 4:1 lipo:AON ratio. After 24 h, treated cells were harvested by centrifugation (3000 rpm, 10 min).

### 2.5. RNA Extraction and cDNA Synthesis

Total RNA was extracted from cultured lymphoblasts and fibroblasts and from a wildtype human foetal retina using the RNeasy Mini Kit (Qiagen, Courtaboeuf, France) according to manufacturer′s protocol. All samples were DNase treated by the RNase-free DNase set (Qiagen, Courtaboeuf, France). Concentration and purity of total RNA was assessed using the Nanodrop-8000 spectrophotometer (Thermo Fisher Scientific, Courtaboeuf, France) before storage at −80 °C. First-stranded cDNA synthesis was performed from 500 ng of total RNA extracted using Verso cDNA kit (Thermo Fisher Scientific, Courtaboeuf, France) with random hexamer:anchored oligo(dT) primers at a 3:1 (voL:voL) ratio according to the manufacturer′s instructions. A non-RT reaction (without enzyme) for one sample was prepared to serve as a control in RTq-PCR and RT-PCR experiments.

### 2.6. RT-qPCR Analysis

To measure the level of expression of *RPGRIP1* mRNA in wildtype fibroblasts, lymphoblasts and in human foetal retina, cDNAs (5 µL of a 1:25 dilution in nuclease-free water) were subjected to real-time PCR amplification in a buffer (20 µL) containing SYBR^®^ Green PCR Master Mix (Applied Biosystems, Thermo Fisher Scientific, Courtaboeuf, France) and 300 nmoL/Lof primers *RPGRIP1 (exonic11F)* forward, 5′-gcataaacaggaagtagagctcctc-3′ and *RPGRIP1 (exonic12R)* reverse, 5′-tggtctctgcgtgtgatacttgca-3′ on a Master cycler (Eppendorf, Montesson, France). Regions within the human β-glucuronidase mRNA (*GUSB*, NM_000181.3), the human hypoxanthine phosphoribosyl transferase 1 mRNA (*HPRT1*, NM_000194), and the human P0 large ribosomal protein mRNA (*RPLP0*, NM_001002.3) were used for normalisation, respectively. Primers and PCR conditions are available in Gerard et al [15]. Data were analysed using the SDS 2.3 software (Applied Biosystems, Thermo Fisher Scientific, Courtaboeuf, France). For each cDNA sample, the mean of quantification cycle (Cq) values was calculated from triplicates (SD < 0.5 Cq). *RPGRIP1* expression levels were normalised to the “normalisation factor” obtained from the geNorm software for Microsoft Excel [16] which uses the most stable reference genes. No reverse transcriptase (non-RT), no template control (NTC) reactions were used as negative controls in each run (Cq values NTC = undetermined, non-RT > 40 and ALBh > 40). The quantitative data are presented as a ratio among values for individual mRNAs. 

### 2.7. RT-PCR Analysis

cDNAs (5 µL) from controls and patients were amplified in 50 µL of 1 × Phusion HF buffer containing 5 mmoL/LdNTPs (Thermo Fisher Scientific, Courtaboeuf, France), 0.02 units of Phusion High-Fidelity DNA polymerase (Thermo Fisher Scientific, Courtaboeuf, France), and 10 µmoL/L of each *RPGRIP1(exonic11F)* and *RPGRIP1(exonic12R)* primers. PCRs were carried out on a 2720 Thermal Cycler (Applied Biosystems, Thermo Fisher Scientific, Courtaboeuf, France) under the following conditions: initial denaturation at 98 °C for 5 min, followed by 30 cycles of 10 sec denaturation at 98 °C, 30 s annealing at 60 °C and 30 s extension at 72 °C. The PCR products were separated (20 µL) by electrophoresis in a 3% agarose gel stained with ethidium bromide and visualised under UV lights. No template control reactions were used as a negative control. The identity of PCR products was determined by Sanger sequencing.

### 2.8. Sanger Sequencing

RT-PCR products from control and patient individuals were separated by electrophoresis onto a 1.5% low-melting agarose gel, cut out and diluted in 1 volume of water at 65 °C. An aliquot (2 µL) was Sanger sequenced using the *RPGRIP1 (exonic11F)* or *RPGRIP1 (exonic12R)* primers and the BigDye^®^ Terminator v3.1 on an ABI3700 sequencer. Data were analysed using the Sequencing Analysis 6 Software (Applied Biosystems, Thermo Fisher Scientific, Courtaboeuf, France).

The recurrence of the c.1468-128T>G variant among unresolved LCA individuals was determined by PCR amplification of genomic DNA (100 ng) in 2 µL of 5× buffer, 1.5 mM of MgCl_2_, 1.25 units of GoTaq^®^ DNA polymerase (Promega, Charbonnières-les-bains, France), 1 mmoL/LdNTPs (Thermo Fisher Scientific, Courtaboeuf, France), and 10 µmoL/Lof each primer *RPGRIP 1(genomic12F)* forward, 5′-gatgaggacttgaaaagatc-3′ and *RPGRIP1 (genomic12R)* reverse, 5′-tcctggtttttgggttcact-3′. PCRs were carried out on a 2720 Thermal Cycler (Applied Biosystems, Thermo Fisher Scientific, Courtaboeuf, France) under the following conditions: initial denaturation at 98 °C for 5 min, followed by 30 cycles of 10 s denaturation at 98 °C, 30 s annealing at 53 °C and 30 s extension at 72 °C. 2 µL were Sanger sequenced using the *RPGRIP1 (genomic12F)*, or *RPGRIP1 (genomic12R)* primers as described above.

### 2.9. Haplotype Analysis and Datation of the c.1468-128T>G Variant

Genetic microsatellite markers flanking the *RPGRIP1* locus were studied for linkage disequilibrium in individuals LCA215, LCA454 LCA426 and MON035 carrying the *RPGRIP1* c.1468-128T>G variant and their parents (LCA485 and LCA903 were excluded due to unavailable parental DNA). The position of the markers (locus name and position in parentheses) and of the mutation were estimated from the human genome GRCh37/hg19 assembly available from the University of California, Santa Cruz (UCSC): AFM199ZF4 (D14S72: 21,370,988–21,371,351)—AFMB329WE5 (D14S1023: 21,441,901–21,442,220)—*RPGRIP1* (c.1468-128G>T: 21,789,290)—AFMA086WB5(D14S1070: 21,548,252–21,548,590)—CHLC.GATA74E02 (D14S742: 22,200,886–22,201,487)—AFM312XH1 (D14S283: 22,687,415–22,687,784).

Amplified markers were electrophoresed on the ABI 3500XL genetic analyser (Applied Biosystems, Thermo Fisher Scientific, Courtaboeuf, France) and analysed using the Fragment Analysis Assay and Genotyper 5 Softwares. For each marker, the heterozygote frequency and the size range of alleles were obtained from the Généthon Linkage Map [17] available from the Centre d’Etude du Polymorphisme Human (CEPH). A founder effect was assessed from the haplotypes using the ESTIAGE software [18].

### 2.10. Immunofluorescence Microscopy

Fibroblasts were seeded at 2.5 × 10^5^ cells/well on glass coverslips in 12-well plates. After 48h of starvation, cells were fixed in ice-cold methanol (5 min at −20 °C) and washed twice in PBS. Cells were permeabilised in PBS supplemented with 3% bovine serum albumin and 0.1% Triton for 1 h at room temperature before being incubated overnight at 4 °C in permeabilisation buffer containing (rabbit anti-pericentrin (1:1000), mouse monoclonal anti-acetylated tubulin (1:1000); Sigma-Aldrich, Saint-Quentin Fallavier, France) primary antibodies. After three washes with PBS, cells were incubated for 1 h at room temperature in permeabilisation buffer containing secondary antibodies (Alexa-Fluor 594– and Alexa-Fluor 488– conjugated goat anti-rabbit IgG (1:1000) and goat anti-mouse IgG (1:1000); Thermo Fisher Scientific, Courtaboeuf, France) followed by three washes with PBS. A mounting medium containing 4′,6- diamidino-2-phenylindole (DAPI) (Prolong Gold, ThermoFisher Scientific Courtaboeuf, France) was used to label nuclei. Immunofluorescence images were obtained using a Spinning Disk Zeiss microscope (Zeiss, Oberkochen, Germany). Final images were generated using ImageJ (National Institutes of Health, Bethesda, MA, USA). The percentage of ciliated cells was calculated from two independent experiments (*n* > 100 cells for each cell line). 

## 3. Results

### 3.1. Identification of Non-Coding RPGRIP1 Alleles

Panel-based analysis of coding sequences and intron–exon boundaries of genes involved in LCA and differential diagnoses in 722 probands revealed biallelic *RPGRIP1* pathologic variants in 26/722 individuals and single presumably loss-of-function *RPGRIP1* variants (one nonsense, one 1-bp duplication, one 1-bp deletion and three consensus splice-site pathologic variants; Table 1) in 6/722 of them. The presence of copy number variations was excluded in all cases. Subsequent case-parent trio-based WLS in these families in search for non-coding and structural variants on the second *RPGRIP1* allele identified the ultra-rare deep intronic c.1468-128T>G (rs1245948143) change in 3/6 families (Table 1, Figure 1, Families: LCA215, LCA454 and MON035). In silico analysis used to assess the impact of this variant lying in intron 11 suggested the creation of a new donor splice-site 5-bp upstream of the change (Splice AI donor gain of 0.64 and SpliceSiteFinder-like, MatEntScan, NNSLPICE, and Human Splicing Finder splice site scores of 79.3, 6.5, 0.8 and 86.2, respectively; Figure 2A). RT-PCR performed on RNA isolated from lymphoblast and fibroblast cell lines of a compound heterozygous (LCA215 with the c.1502_1505dup) individual and fibroblasts from a homozygous individual (LCA426) using a primer pair designed to amplify *RPGRIP1* exons 11 and 12, detected the wildtype mRNA and a higher molecular weight product in both lines (194-bp and 312-bp fragments; Figure 2C). Sequencing of the aberrant 312-bp product revealed the insertion of a 118-bp cryptic exon between exons 11 and 12, which introduces a stop codon downstream of exon 11 (r.1467_1468ins1468-250_1468-133; Figure 2B). The aberrant splice product was not detected in two control individuals (Figure 2C). In silico analysis of the intron 11 sequence detected a cryptic acceptor splice-site 1-bp upstream of the 5’ extremity of the cryptic exon (MatEntScan, NNSLPICE, GeneSplicer and Human Splicing Finder splice site scores of 4.1, 0.5, 1.1 and 86.1, respectively). Interestingly, a more upstream cryptic acceptor splice site with higher splicing scores (SpliceSiteFinder-like, MatEntScan, NNSLPICE, GeneSplicer and Human Splicing Finder splice site scores of 83.9, 10.7, 0.9, 6.1 and 88.3, respectively) was identified which was not used by the splicing machinery either in lymphoblasts or fibroblasts (Figure 2A).

To determine whether this variant could be a common cause of LCA, we screened the 106/722 unresolved unrelated index cases of our cohort by Sanger sequencing. This allowed the identification of the variant in three unrelated individuals, two of whom were homozygote (LCA426, LCA903) and one was heterozygote (LCA485). *RPGRIP1* WLS in this latter individual identified a second deep intronic variant in intron 7 (c.930+77A>G), predicted to create donor and acceptor splice sites (SpliceSiteFinder-like, MaxEntScan, NNSLPICE and Human Splicing finder splice scores of 82.0, 8.9, 0.9, 85.7 and 88.8, 8.5, 0.9, 92.6, respectively; Appendix A). RNA was not analysed but it is very likely that splicing is affected. The variant could lead to the retention of the c.630+1_630+77 (r.930_931ins630+1_630+77) and/or the c.630+78_631-1 (r.930-931ins630+78_631-1) intronic sequences (Appendix A). Familial segregation analysis confirmed biparental transmission of the c.1468-128T>G in LCA426 homozygous individual. Whether the c.930+77A>G variant occurred in trans of the c.1468-128T>G allele in LCA485 could not be determined due to the absence of parental DNA samples.

Analysis of patient haplotypes harbouring the c.1468-128T>G variant (5/6, LCA485 excluded due to inability to determine the genetic phase) using the ESTIAGE software identified patients LCA215 et LCA454 as sharing the largest common haplotype (758 Kb), giving an estimate of the age of the founder haplotype of 68 generations (IC95 = (28; 271)) assuming a mutation rate of 10^−6^ (Table 2).

### 3.2. Approach to Repair the c.1468−128T>G Change

We assessed the efficacy of AONs to redirect the splicing to the consensus splice sites in cells from one patient carrying the c.1468−128T>G variant (LCA215). The transfection of the lymphocytes from LCA215 for 24 h using 150 nmoL/L of AON1, 2 or 3 but not the control AONsense oligonucleotide successfully hampered the use of the cryptic splice sites introduced by the deep intronic variant as determined by RT-PCR analysis. Neither the AONs nor the AONsense had apparent effect on *RPGRIP1* in control cells. Given the role of *RPGRIP1* in ciliogenesis, we measured the abundance of its mRNA in control fibroblasts and showed extremely low levels compared to human foetal retina (0.005 times) and even lower levels than in lymphoblasts (Appendix A). Consistent with a minor endogenous *RPGRIP1* expression in fibroblasts, the abundance primary cilia upon serum-starvation and axonemal length were not affected in fibroblasts from patients LCA215 and LCA426 compared to controls (85% of ciliated cells, Appendix A and mean axonemal sizes of 3.5 m in controls or patients, Appendix A). 

### 3.3. RPGRIP1-Associated Retinal Disease

Review of the natural history of the disease and ophthalmological data in individuals carrying deep intronic *RPGRIP1* pathologic variants in homozygosity or compound heterozygosity presented with nystagmus, oculo-digital signs of Franceschetti, photophobia, hyperopia and a vision function from light perception to at best 10/200. This disease presentation is consistent with previous reports [19,20,21].

In 3/6 families carrying single truncating *RPGRIP1* variants, WLS failed to detect a rare variant likely to alter splicing or expression on the second allele (Families: LCA489, LCA657, MON017; Table 1). LCA489 presented a severe and early form of retinal dystrophy diagnosed at 10 months with nystagmus, oculo-digital signs and an extinguished ERG. Her parents described improvement during her childhood with night blindness and a concentric reduction in her visual field. Today, she has high hyperopia + 10 (LRE), a visual acuity of 20/125 LRE. LCA657 was described as severe cone–rod dystrophy associated with hearing loss and mild intellectual disability. Recently, he had a son with exactly the same phenotype redirecting the mode of transmission in autosomal dominant. MON017 presented a severe and early retinal dystrophy with hyperopia +4 (LRE), visual acuity of 20/50 LE and 20/40 RE, and extinguished ERG (Table 1).

## 4. Discussion

With the advent of cost-effective high throughput sequencing, disease-causing mutations in non-coding regions are being increasingly disclosed. With respect to IDRs, founder deep intronic mutations have been reported in *ABCA4* and *CEP290* which turned out to be major causes of Stargardt disease and LCA, respectively [22,23,24,25,26]. Here, we report the results of a genetic study designed to increase the diagnosis yield of LCA. In this aim, we combined trio-based WLS in individuals carrying one single *RPGRIP1* pathologic variant with Sanger sequencing analysis of non-coding changes in all LCA cases with negative molecular test results. These individuals were selected through gene-panel molecular diagnostics of a cohort of 722 probands addressed for a retinal disease suggestive of LCA, which identified biallelic and monoallelic unequivocal *RPGRIP1* pathologic variants in 26/722 and 6/722 individuals, respectively. Starting with the six probands, we were able to identify nine additional variant *RPGRIP1* alleles in 3/6 of them and 3/106 unresolved cases, increasing the contribution of *RPGRIP1* variant from 26/722 (3.6%) to 32/722 (4.4%). This is significant and likely underestimated, assuming the low frequency of *RPGRIP1* pathologic variants among LCA cases and the bias of selecting cases with one loss-of-function variant, respectively. In total, panel-based sequencing and WLS detected 60 *RPGRIP1* alleles carrying pathologic variants, 8 of which consisted in the c.1468-128T>G deep intronic variation identified in six families (8/60, 13% of *RPGRIP1* mutant alleles; 6/32, 18.75% of families with biallelic *RPGRIP1* pathologic variants). This change is the most frequent pathologic *RPGRIP1* variant in our cohort. It was identified in apparently unrelated families. However, upon investigation they were found to originate from Western France and haplotype analysis suggested an ancient founder effect, further supported by homozygosity in one non-consanguineous family.

Consistent with the view that LCA combines the earliest and most severe rod–cone and cone–rod dystrophies, some milder LCA gene mutations have been reported in less severe IRDs. These include *RPGRIP1*, a pathologic variant which has been reported to cause an early-onset cone–rod dystrophy with loss of vision in teenage (CORD13, MIM608194; [27]). Individuals carrying the c.1468-128T>G in homozygous or in heterozygous state with a loss-of-function variant or the other deep intronic c.930+77A>G change, displayed a typical *RPGRIP1*-associated LCA disease [19,20,21]. This observation suggests that both the c.1468-128T>G and c.930+77A>G variants have a deleterious impact on photoreceptor connecting cilium where RPGRIP1 localises. Loss of expression in the Rpgrip1^nmf247^ mouse which recapitulates well the clinical expression of LCA causes shortening of photoreceptor outer segments [28]. Consistent with a basal expression of RPGRIP1, fibroblasts from this mouse [29] such as those from the patients carrying the c.1468-128T>G in homozygosity or compound heterozygosity revealed apparently normal ciliogenesis upon serum-starvation, as determined by cilia abundance and axonemal length measurements.

The c.1468-128T>G variation (as the c.930+77A>G change) is predicted to introduce a pseudo-exon PE encoding a premature termination codon in some of the *RPGRIP1* mRNA, while allowing the production of the wildtype product. Analysis of fibroblast mRNA from an individual carrying the c.1468−128T>G change in homozygosity (LCA426) confirmed this prediction. Whether the wildtype mRNA is transcribed in photoreceptors as well is likely. Given the severity of the disease it seems that the wildtype mRNA transcribed from the mutant allele together with a mutant mRNA is insufficient to compensate for the *RPGRIP1* deficit.

Significant progress has been made utilising gene augmentation therapy for a few genetic subtypes of IRD [30]. *RPGRIP1* is an attractive candidate for this approach given the preservation of foveal photoreceptors despite severely impaired visual function in young patients [31] and the already established success in murine [32] and canine [33] models. Antisense oligonucleotide (AON)-mediated splice modulation which is a powerful method to correct the consequences of mutations that affect pre-mRNA splicing, is another attractive method to treat *RPGRIP1* deep intronic variants. It has demonstrated promising results in clinical trials for several inherited disorders [34]. This includes LCA10 cases due to the *CEP290* c.2991+1655A>G mutation, another founder deep intronic mutation which gives rise to an aberrant product containing a frameshifting PE and the wildtype mRNA and alters ciliation in photoreceptors and fibroblasts [15,34,35,36,37]. Furthermore, AONs have been proven effective in redirecting the splicing machinery towards the consensus splice sites in primary fibroblasts [15], IPSC-derived 3D retinal organoids [37] and humanised mice carrying the mutation [38] and ultimately in patients who long retain central photoreceptors amenable to treatment [39]. In this light, the most frequent *RPGRIP1* variation c.1468-128T>G is a very attractive candidate for AON-mediated splice-switching therapeutic developments. In the absence of a functional readout in fibroblasts which could be used to make proof-of-concept of AON-mediated splice-switching therapy, we used patient lymphoblasts due to higher RPGRIP1 mRNA abundance. Using AONs targeting the acceptor site (AON1), an exonic splice enhancer (ESE, AON2) and the donor site (AON3), we were able to prevent the use of the mutant splice site, paving the way to further AON-mediated therapy.

*RPGRIP1* WLS failed to detect a second disease allele in 3/6 individuals in whom panel-based molecular diagnosis identified loss-of-function variants on one *RPGRIP1* allele. Two of them presented a severe cone rod dystrophy with the clinical features characteristic for *RPGRIP1*-associated retinal degeneration. The third family seems to harbour the first heterozygous variant by chance and the recent knowledge of an affected son changes the search for a gene with the dominant mode of heredity. Whether a long-distance mutation exists which affects RPGRIP1 expression or whether another gene is responsible for the disease in any of these cases is possible. Whole genome sequencing will hopefully help to address this question.

In conclusion, we show that, so far, the most common *RPGRIP1* disease allele in France consists of a deep intronic variant that is amenable to AON-mediated therapy. This observation corroborates previous work suggesting that noncoding *RPGRIP1* variants are not uncommon in the pathogenesis of recessive IRDs [40] and highlights the need to include non-coding regions of known genes in routine molecular testing.

## Figures and Tables

**Figure 1 genes-12-00287-f001:**
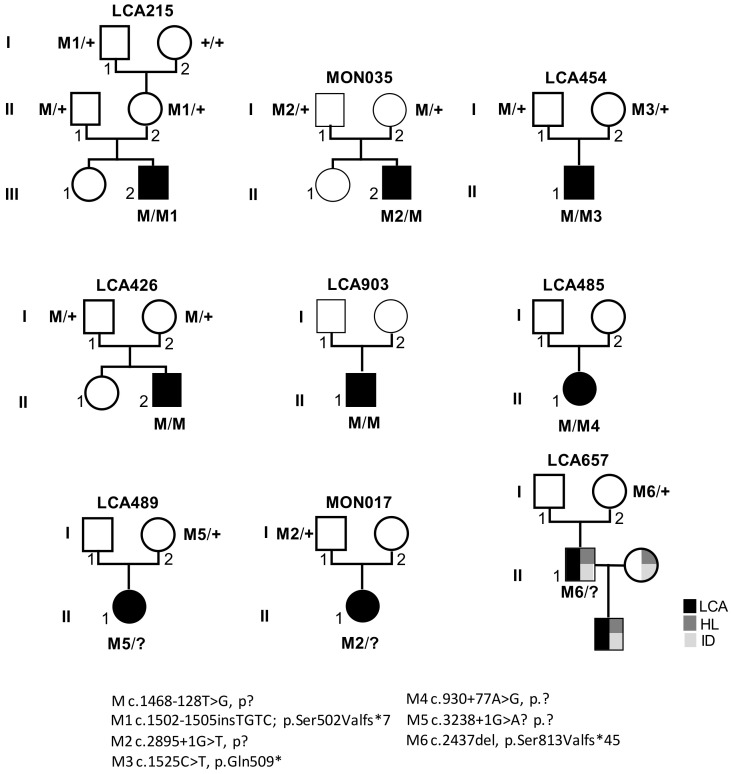
Pedigrees of LCA families and segregation analysis of *RPGRIP1* pathologic variants. M: c.1468-128T>G; p.?, M1 c.1502-1505insTGTC; p.Ser502Valfs*7, M2 c.2895+1G>T, p?, M3 c.1525C>T, p.Gln509*, M4 c.930+77A>G, p.?, M5 c.3238+1G>A? p.?, M6 c.2437del, p.Ser813Valfs*45; +: wild-type allele. LCA: Leber congenital amaurosis, HL: hearing loss, ID: intellectual disability.

**Figure 2 genes-12-00287-f002:**
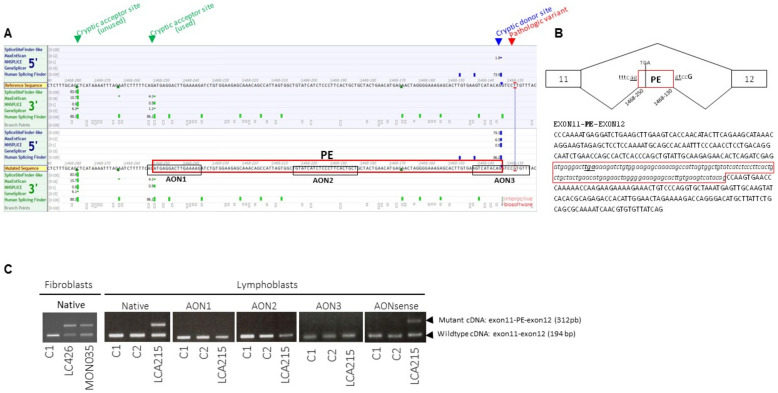
Splicing prediction scores around the *RPGRIP1* c.1468-128T>G variant in intron 11, position and sequence of the pseudo-exon amplified from patient mRNA and position of splice-switching antisense oligonucleotides (AONs). (**A**) Representation of the intron 11 region encompassing the c.1468-128T>G variant and cryptic acceptor and donor splice-sites with splicing score predictions according to the Alamut Software. The pseudo-exon (PE) is framed in red. The sequences of the three AONs designed to target the cryptic acceptor splice site (AON1), one exonic splice enhancer (ESE) sequence in the cryptic exon (AON2) and the cryptic donor splice site (AON3) designed using the ESEFinder 3.0 program are framed in black.(**B**) Schematic representation of wild-type and mutant RPGRIP1 mRNAs produced from the c.1468-128T>G mutant allele in both homozygous and heterozygous individuals and sequence of the mutant mRNA encompassing the pseudo-exon (PE, framed in red) between exons 11 and 12. (**C**) RT-PCR from mRNA isolated from controls (C1, C2) and patients carrying the c.1468-128T>G variant in homozygosity (LCA426) and compound heterozygosity (MON035, LCA215) in native condition and following treatment of C1, C2 and LCA215 lymphoblasts with 150 nmoL/L of AON1, AON2, AON3 or AONsense. In native conditions, RT-PCR from mRNA extracted from LCA426 and MON035 fibroblasts and LCA215 lymphoblasts yielded two products, the wildtype product (exon 11–12: 194 bp) and a higher molecular weight fragment (exon 11-PE-12: 312 bp). The controls C1 and C2 only display the wildtype 194-bp product. The treatment of LCA215 lymphoblasts with AON1, AON2, or AON3, but not the negative control sense oligonucleotide (AONsense), resulted in the disappearance of the higher molecular weight product encompassing the aberrant pseudo-exon.

**Table 1 genes-12-00287-t001:** *RPGRIP1* variants and clinical characteristics of the patients with Leber congenital amaurosis (LCA) included in the present study. LP: light perception, RLE: right left eye, ERG: electroretinogram, na: not available.

		Pathologic Variants				Clinical Features at the Diagnosis							Extraocular Features
Patient	Gender	Allele 1		Allele 2		Nystagmus	Oculo-Digitals Signs	Photophobia	Refraction (Age in Years)	Visual Acuity	Fundus	ERG	
		cDNA change	Protein change	cDNA change	Protein change								
**LCA215**	M	c.1502_1505dup	p.Gln503Valfs*6	c.1468-128T>G (r.1467_1468ins1468-250_1468-133)	p.?	Yes	Yes	No	+ 2.5LRE (27)	Reduced to LP	Salt and pepper aspect	Undetectable	No
**LCA454**	M	c.1525C>T (rs61751267)	p.Gln509*	c.1468-128T>G (r.1467_1468ins1468-250_1468-133)	p.?	Yes	Yes	No	+ 7.5LRE (2)	Reduced to LP		Undetectable	No
**MON035**	M	c.2895+1G>T (rs748072501) (r.spl)	p.?	c.1468-128T>G (r.1467_1468ins1468-250_1468-133)	p.?	Yes	No	No	+ 6.25RE; + 4.75LE (0.5)	Reduced to hand movement	Reduced calibers of the vessels	Undetectable	No
**LCA426**	M	c.1468-128T>G (r.1467_1468ins1468-250_1468-133)	p.?	c.1468-128T>G (r.1467_1468ins1468-250_1468-133)	p.?	Yes	No	Yes	+ 4LRE (6)	Reduced to LP	Salt and pepper aspect	Undetectable	No
**LCA903**	M	c.1468-128T>G (r.1467_1468ins1468-250_1468-133)	p.?	c.1468-128T>G (r.1467_1468ins1468-250_1468-133)	p.?	Yes	Yes	No	+ 1LRE (39)	Reduced to LP	Reduced calibers of the vessels, salt and pepper aspect	Undetectable	No
**LCA485**	F	c.930+77A>G (r.spl)	p.?	c.1468-128T>G (r.1467_1468ins1468-250_1468-133)	p.?	na	na	na	na	na	na	Undetectable	No
**LCA657**	M	c.2437del	p.Ser813Valfs*45	-	-	Yes	No	na	na (22)	20/500 LRE	Reduced calibers of the vessels	Undetectable	Transmission of hearing loss and mild intellectual deficiency
**LCA489**	F	c.3238+1G>A (r.spl)	p.?	-	-	Yes	Yes	No	+ 10LRE (14)	20/125LRE	Salt and pepper aspect	Undetectable	No
**MON017**	F	c.2895+1G>T (r.spl)	p.?	-	-	Yes	No	Yes	+ 4LRE (3)	20/50LE20/40RE	Salt and pepper aspect	Undetectable	No

**Table 2 genes-12-00287-t002:** Linkage analysis of the *RPGRIP1* region using microsatellites D14S72, D14S1023, D14S1070, D14S742 and D14S283 in four independent families harbouring the c.1468-128T>G. The common haplotype is in red.

	Genomic Position	MON035		LCA426		LCA454		LCA215		Frequency of the Allele in Cis of the Mutation
D14S72	21,370,988	254	266	254	254	254	258	254	266	0.167
D14S1023	21,441,901	94	89	94	94	94	89	96	91	0.232
D14S1070	21,548,252	244	258	244	244	244	260	244	260	0.304
MUTATION	21,789,290	1	0	1	1	1	0	1	0	0
D14S742	22,200,886	396	396	396	396	396	400	400	404	0.296
D14S283	22,687,415	129	135	129	129	129	140	121	121	0.143

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
