# Peer review of "Whole Locus Sequencing Identifies a Prevalent Founder Deep Intronic RPGRIP1 Pathologic Variant in the French Leber Congenital Amaurosis Cohort"

_genes, 2021, doi:10.3390/genes12020287_

Round 1
Reviewer 1 Report
This paper describes the identification of two (deep) intronic variants in the RPGRIP1 gene and their contribution to LCA. One of both was shown to be a founder variant. In addition, AONs were used to reverse the splice defect the latter. Overall, this study provides an important contribution to the IRD field.
I have the following (minor) suggestions:
- Title: whole gene sequencing: rather whole locus resequencing as you included upstream and downstream regions as well?
- The employed strategy, consisting of locus resequencing in mono-allelic cases, is quite elegant compared to whole genome studies, which are in these cases a bit overkill. On the other hand, custom probe designs like this can be quite expensive for small patient groups. Are they still more cost-efficient compared to less expensive whole genome sequencing preps (but with a higher sequencing cost)? What would be your advice for other autosomal recessive IRD genes with frequent reports of single heterozygous variants?
- Line 93: how were these 55 genes selected? Could you list these genes?
- Lines 114 and 115: which frequency threshold was used to filter out common polymorphisms?
- I might have overlooked this in the paper, but could you add the reference transcript (eg RefSeq NM_ number or Ensembl ENST number) that was used for the RPGRIP1 mutation nomenclature?
- Lines 261-262: “In silico analysis used to assess the impact of this variant lying in intron 11 suggested the creation of a new donor splice-site 5-bp downstream of the change”: is this correct, or do you mean upstream? If the prediction was downstream of the mutation, it might be interesting to include this prediction in Figure 2A. Now, Figure 2A only shows donor splice site predictions upstream of the mutation.
- Lines 264-268: were the lymphoblast and fibroblast cell lines treated with an agent that blocks nonsense-mediated decay (such as puromycin or cycloheximide)? The different bands in these cell lines that are described in the text are not visible in Figure 2C (“native”). In general, this figure is not clear, it seems as if it is a cropped version of a bigger gel image; could you show the full image? Also, does this figure show the result of RT-PCR of RT-qPCR experiments?
- Lines 282-283: could you provide more information on this variant, eg show a figure similar to Figure 2A? Do you think a new PE will be formed, or will this variant rather result in intron retention because of the predictions and quite close proximity to the previous exon? Is a lymphoblast cell line available from the proband for cDNA sequencing?
- Line 339: As you do not include the percentage of cases solved by other LCA genes, I'm not sure if it is correct to state these numbers as “diagnosis yield”. They rather refer to the contribution of RPGRIP1 mutations to your LCA patient cohort.
Typo’s:
- Line 33: of a donor splice-site: remove of
- Line 34: which we able to hamper: we were able to hamper
- Line 34: antisens > antisense
- Line 79: and show correction the abnormal splicing > correction of
- Lines 177 and 178: twice the same sentence (Primer sequences and RTq-PCR conditions available in Gerard et al)?
- Line 208: de
- Lines 330-333: weird sentence
- Lines 346: the all of them
- Line 348: families > family
- Line 360: from > from
- Line 373: preservation of foveal photoreceptors?
- Line 386: introduces of a PE
Author Response
Reply to Reviewer 1
We are very grateful to the Reviewer for her/his positive comments and his kind suggestions, which we have taken into account in the revised version of the manuscript.
Point by point answer to the Reviewer 1 comments.
- Title: whole gene sequencing: rather whole locus resequencing as you included upstream and downstream regions as well?
The title has been modified as indeed some non-coding regions have been included.
- The employed strategy, consisting of locus resequencing in mono-allelic cases, is quite elegant compared to whole genome studies, which are in these cases a bit overkill. On the other hand, custom probe designs like this can be quite expensive for small patient groups. Are they still more cost-efficient compared to less expensive whole genome sequencing preps (but with a higher sequencing cost)? What would be your advice for other autosomal recessive IRD genes with frequent reports of single heterozygous variants?
We thank the reviewer for drawing attention on cost issues which are indeed of major importance when considering molecular diagnosis strategies.
Whole genome sequencing at our Institute is charged 1,000 Euros per individual. The cost to resolve the molecular diagnosis of the individuals we report in the ms was 460 euros, which can be decomposed as:
-Index case 1st intention panel sequencing: Probes 60 euros (3,000 euros/50 individuals) & Sequencing 100 euros.
-Trio whole-sequencing panel (14 LCA genes): Probes 60 euros & sequencing 300 euros.
Based on the success rate in RPGRIP1 mono-allelic cases and some other genes we have now included probes that cover the locus of the 20 LCA genes referenced in the OMIM database as part of the 1st intention panel. With this panel the cost to solve the RPGRIP1 cases would have been 160 euros (index case sequencing). We think that whole locus sequencing which provides high sequencing coverage and depth is preferable to WGS in some recognizable IRDs with genetic homogeneity (e.g. Stargardt diseases) or somewhat limited genetic heterogeneity (e.g. Best disease and other macular dystrophies, CSNB, ACHM, LCA, Albinism etc.). When considering rod-cone and cone-rod dystrophies in general, the genetic heterogeneity is so high that whole locus sequencing may not be cost-effective compared to WGS.
- Line 93: how were these 55 genes selected? Could you list these genes?
The list of the genes and the rational of their selection have been included in a supplementary table (Supplementary Table 1): “A custom panel was designed to include genes involved in LCA, EOSRD, differential diagnosis and ciliopathies and neurometabolic diseases which can manifest initially as an LCA or an EOSRD according to the OMIM database or the literature (at least one reported case). The target regions encompassed 468 kb, covered 966 exons, 1000 bases of 5' and 3' UTRs and 50bp spanning splice junctions of 55 genes.”
- Lines 114 and 115: which frequency threshold was used to filter out common polymorphisms?
Considering the rarity of the disease, the recessive mode of inheritance and the genetic heterogeneity, we assumed that the affected individuals were compound heterozygote with a maternal/paternal already known mutation and for a second variant with a minor allele frequency < 0.01 in databases. And thus excluded all variants with a MAF≥ 0.01. The frequency has been added in the revised versions of the ms: “Genome variations were defined using the in-house software PolyDiag for gene-panel analysis, which filters out irrelevant and common polymorphisms based on frequencies (minor allele frequency ≥ 0.01) extracted from the Genome Aggregation Database (gnomAD; https://gnomad.broadinstitute.org/)...”.
- I might have overlooked this in the paper, but could you add the reference transcript (eg RefSeq NM_ number or Ensembl ENST number) that was used for the RPGRIP1 mutation nomenclature?
We are sorry to have forgotten this important information. The RefSeq number (NM_020366.3) has been added in the revised version.
- Lines 261-262: “In silico analysis used to assess the impact of this variant lying in intron 11 suggested the creation of a new donor splice-site 5-bp downstream of the change”: is this correct, or do you mean upstream? If the prediction was downstream of the mutation, it might be interesting to include this prediction in Figure 2A. Now, Figure 2A only shows donor splice site predictions upstream of the mutation.
Thank you for pointing this inconsistency. Indeed, the new donor splice-site lies 5-bp upstream of the change. This has been corrected in the revised version of the ms.
- Lines 264-268: were the lymphoblast and fibroblast cell lines treated with an agent that blocks nonsense-mediated decay (such as puromycin or cycloheximide)?
It would have been interesting to use an NMD inhibitor but unfortunately we did not in this study.
- The different bands in these cell lines that are described in the text are not visible in Figure 2C (“native”). In general, this figure is not clear, it seems as if it is a cropped version of a bigger gel image; could you show the full image? Also, does this figure show the result of RT-PCR of RT-qPCR experiments?
Figure 2C shows the result of RT-PCR. We can observe that the patients LCA426, MON035 and LCA215 display two fragments, the wildtype product (exon 11-12: 194 bp) and a higher molecular weight fragment (exon 11-PE-12: 312 bp). The controls C1 and C2 only display the wildtype 194 bp product. The arrows on the full images of the gels shown below indicate the tracks which have been used to prepare Figure 2C (black: original version; red: revised version). We have not used the full images in the manuscript because we had to reorganize the panels for the sake of order of presentation native, AON1, 2, 3 and sense.
We have added some annotations on the Figure to clarify the description and modified the legend of Figure 2C: (C) RT-PCR from mRNA isolated from controls (C1, C2) and patients carrying the c.1468-128T>G variant in homozygosity (LCA426) and compound heterozygosity (MON035, LCA215) in native condition and following treatment of C1, C2 and LCA215 lymphoblasts with 150 nmol/l of AON1, AON2, AON3 or AONsense. In native conditions, RT-PCR from mRNA extracted from LCA426 and MON035 fibroblasts and LCA215 lymphoblasts yielded two products, the wildtype product (exon 11-12: 194 bp) and a higher molecular weight fragment (exon 11-PE-12: 312 bp). The controls C1 and C2 only display the wildtype 194 bp product. The treatment of LCA215 lymphoblasts wih AON1, AON2, or AON3, but not the negative control sense oligonucleotide (AONsense), resulted in the disparition of the higher molecualr weight product encompassing the abberant pseudo-exon.
- Lines 282-283: could you provide more information on this variant, eg show a figure similar to Figure 2A? Do you think a new PE will be formed, or will this variant rather result in intron retention because of the predictions and quite close proximity to the previous exon? Is a lymphoblast cell line available from the proband for cDNA sequencing?
We have modified the text and added a supplementary figure (Supplementary Figure 1) which shows that the c.930+77A>G variant is predicted to activate cryptic donor and acceptor splice sequences at the site of the mutation, predicting the retention of the 77-first nucleotides and/or of the 1139-last nucleotides of intron 7 into the mRNA: “This allowed the identification of the variant in three unrelated individuals, two of whom were homozygote (LCA426, LCA903) and one was heterozygote (LCA485). RPGRIP1 WLS in this latter individual identified a second deep intronic variant in intron 7 (c.930+77A>G), predicted to create a donor and an acceptor splice sites (SpliceSiteFinder-like, MaxEntScan, NNSLPICE and Human Splicing finder splice scores of 82.0, 8.9, 0.9, 85.7 and 88.8, 8.5, 0.9, 92.6, respectively; Supplementary Figure 1). RNA has not been analysed but it is very likely that splicing is affected. The variant could lead to the retention of the c.630+1_630+77 (r.930_931ins630+1_630+77) and/or the c.630+78_631-1 (r.930-931ins630+78_631-1) intronic sequences (Supplementary Figure 1).”
- Line 339: As you do not include the percentage of cases solved by other LCA genes, I'm not sure if it is correct to state these numbers as “diagnosis yield”. They rather refer to the contribution of RPGRIP1 mutations to your LCA patient cohort.
To gain in clarity, we have modified the discussion as following: “…we were able to identify nine additional variant RPGRIP1 alleles in 3/6 of them and 3/106 unresolved cases, increasing the contribution of RPGRIP1 variant from 26/722 (3.6%) to 32/722 (4.4%).”
Typo’s:
- Line 33: of a donor splice-site: remove of
“of” has been removed
- Line 34: which we able to hamper: we were able to hamper
Done
- Line 34: antisens > antisense
Done
- Line 79: and show correction the abnormal splicing > correction of
The sentence including this text has been rephrased upon the suggesting of Reviewer 2.
- Lines 177 and 178: twice the same sentence (Primer sequences and RTq-PCR conditions available in Gerard et al)?
Sorry for the typo. One of the sentences has been deleted.
- Line 208: de
It has been replace by “of”
- Lines 330-333: weird sentence
We rephrased as following "Here, we report the results of a genetic study designed to increase the diagnosis yield of LCA. In this aim, we combined trio-based WLS in individuals carrying one single RPGRIP1 pathologic variant with Sanger sequencing analysis of non-coding changes in all LCA cases with negative molecular test results.”
- Lines 346: the all of them
Replaced by “they”
- Line 348: families > family
Done
- Line 360: from > from
“form” has been replaced by “from”
- Line 373: preservation of foveal photoreceptors?
“of” has been added
- Line 386: introduces of a PE
The sentence including this text has been rephrased upon the suggesting of Reviewer 2.
Reviewer 2 Report
A study by Isabelle Perrault et al. (Manuscript ID: genes-1097715) presenting WGS-trio combined with Sanger sequencing as an important method to identify deep intronic variants in RPGRIP1 gene causing LCA in a cohort of patients of French origin.
This manuscript highlights the importance of WGS used as a diagnostic strategy for unsolved cases of advanced retinal degeneration caused by deep intronic variants. This is important because the diagnostic yield of clinically available panel-based NGS methods accounts for 60-70%. The emerging therapies for retinal degeneration and advanced clinical trials emphasize the importance of early genetic diagnosis, which enables eligibility for treatment/or clinical study. Pathogenic variants in RPGRIP1 are a rare cause of LCA.
This study has few main points:
1.Despite its niche focus and study based on French population only, this manuscript represents a large number of patients with LCA-associated with pathogenic variants in RPGRIP1 gene extracted from large cohort of over 700 patients with LCA.
2.The importance of screening for deep intronic variants in unsolved cases is highlighted through the entire manuscript, which gives an extra strength.
- The authors propose the in vitro approach to repair genetic changes as an interesting insight towards the future gene therapy.
4.The methodology is presented adequately.
Weaknesses:
1.Results paragraph: structure need to be improved to highlight the findings better by e.g. dividing the text into sections with subtitles.
Broad comments:
- Change “mutation” to “pathologic variant” in the entire manuscript.
- Nomenclature of pathologic variants should be written according to the latest recommendations of HGVS
- The “Results” paragraph would be more attractive to the reader if divided into sections, e.g. clinical data, genetic information, approach to repair the genetic change.
- Please add ref. to introduction and/or discussion paragraph:
-Retinal structure in Leber's congenital amaurosis caused by RPGRIP1 mutations.
Miyamichi D, Nishina S, Hosono K, Yokoi T, Kurata K, Sato M, Hotta Y, Azuma N.
Hum Genome Var. 2019 Jun 27;6:32. doi: 10.1038/s41439-019-0064-8.
-Progress in treating inherited retinal diseases: Early subretinal gene therapy clinical trials and candidates for future initiatives.
Garafalo AV, Cideciyan AV, Héon E, Sheplock R, Pearson A, WeiYang Yu C, Sumaroka A, Aguirre GD, Jacobson SG. Prog Retin Eye Res. 2020 Jul;77:100827. doi: 10.1016/j.preteyeres.2019.100827.
-Contribution of noncoding pathogenic variants to RPGRIP1-mediated inherited retinal degeneration. Jamshidi F, Place EM, Mehrotra S, Navarro-Gomez D, Maher M, Branham KE, Valkanas E, Cherry TJ, Lek M, MacArthur D, Pierce EA, Bujakowska KM.
Genet Med. 2019 Mar;21(3):694-704. doi: 10.1038/s41436-018-0104-7
Specific comments:
Abstract, row 31: remove bracket after variant symbol c.930+77A>G).
Abstract, row 34: “which we able” – change to “which we were able”.
Introduction, row 43: “manifesting in severe vision impairment or blindness at, or near, birth” – change to “manifesting in severe vision impairment or blindness by one year of age”.
Introduction, row 46-47: change the context since achromatopsia and some forms of CSNB are considered now as slowly progressive.
Introduction, row 57-58: “This, with the advent of therapies for these retinal diseases, gives molecular testing in these diseases an even higher relevance”. Please change this statement to e.g. “The importance of genetic diagnosis in the advent of recently evolving therapies for inherited retinal degenerations is warranted”.
Introduction, row 58-60: please, improve grammar in this statement.
Introduction, row 68-69: please change to e.g. “Since the protein is almost exclusively expressed in the retina, pathologic variants in RPGRIP1 cause a non-syndromic form of LCA”
Introduction, row 78-80: change this statement to e.g. “In our study we made an attempt to AON treatment of the lymphoblasts obtained from patients carrying the change. We were able to correct the abnormal splicing, which could have an important insight into splice-switching therapy of this frequent RPGRIP1 disease allele (13%)”.
Results, row 254: remove this statement “(52 RPGRIP1 disease alleles, article in preparation)”.
Results, row 287: “could not be determined for lack of parental DNA” – please change to “could not be determined due to absence of parental DNA”.
Results, row 301: “compared human fetal retina” – change to “compared to human fetal retina”.
Discussion, row 337: remove “(article in preparation)”.
Discussion, row 353-354: “be it in homozygosity or in compound heterozygosity” – please change to “in homozygous or heterozygous state”.
Discussion, row 373: “preservation foveal photoreceptors” – change to “preservation of foveal photoreceptors”.
Discussion, row 374-376: change to “Antisense oligonucleotide (AON)-mediated splice modulation, which is a powerful method to correct the consequences of pathogenic variants affecting pre-mRNA splicing, is another attractive method to treat RPGRIP1 deep intronic variants”.
Discussion, row 381: change to “Furthermore, antisense oligonucleotides…”
Discussion, row 385-387: please change this statement “In this light, the most frequent RPGRIP1 variation c.1468-128G>T is a very attractive candidate for antisense oligonucleotide (AON)-mediated splice-switching therapeutic developments”.
Discussion, row 396-397: change to: “Two of them presented a severe cone-rod dystrophy with the clinical features characteristic for RPGRIP1-associated retinal degeneration”.
Discussion, row 399: change “exist” to “exists”.
Table 1: improve the table: separate “Gender” from the “cDNA and protein changes”, should appear as a separate column. Write variants according to latest HGVS recommendations. “Extra ocular features” – the table does not have upper border. Consider to change “flat rod and cone ERG” to “undetectable ERG”.
Table 2: improve borders of the table.
Figure 2: text row 502 change to “cryptic”
Figure 2A – is not well readable
Author Response
We are very grateful to the reviewer for her/his positive comments and all the kind suggestions made, we took into consideration in the revised version of the manuscript.
Point by point answer to the Reviewer 2 comments.
Broad comments:
- Change “mutation” to “pathologic variant” in the entire manuscript.
We deleted “mutation” as proposed and had “pathologic variant" or "variant"
Nomenclature of pathologic variants should be written according to the latest recommendations of HGVS
We have now included the description of the c.1468-128C>T and c.930+77A>G pathologic changes at the RNA level: r.1467_1468ins1468-250_1468-133 and r.spl, respectively.
- The “Results” paragraph would be more attractive to the reader if divided into sections, e.g. clinical data, genetic information, approach to repair the genetic change.
Following the advice of the reviewer, sections and headings have been added in the revised version of the manuscript to improve clarity.
Please add ref. to introduction and/or discussion paragraph:
We thank the reviewer for pointing these references which should have been included in the initial version of the ms:
-Miyamichi et al. 2019 has been included as reference 20: "This disease presentation is consistent with previous reports19,20."
-Garafalo et al. 2020 has been included as reference 30: "Significant progress has been made utilizing gene augmentation therapy for a few genetic subtypes of IRD30"
-Jamshidi et al. 2020 has been included as reference 40: “In conclusion, we show that so far the most common RPGRIP1 disease allele in France consists in a deep intronic variant that is amenable to AON-mediated therapy. This observation corroborates previous work suggesting that noncoding RPGRIP1 variants are not uncommon in pathogenesis of recessive IRDs40 and highlights the need to include non-coding regions of known genes in routine molecular testing.
Specific comments:
Abstract, row 31: remove bracket after variant symbol c.930+77A>G).
We have rephrased as following: “We employed Sanger sequencing to search for the pathologic variant in unresolved LCA cases (106/722) and identified three additional families (2 homozygous and 1 compound heterozygous with the NM_020366.3:c.930+77A>G deep intronic change). This makes the c.1468-128T>G the most frequent RPGRIP1 disease allele (8/60, 13%) in our cohort.
Abstract, row 34: “which we able” – change to “which we were able”.
Done
Introduction, row 43: “manifesting in severe vision impairment or blindness at, or near, birth” – change to “manifesting in severe vision impairment or blindness by one year of age”.
Done
Introduction, row 46-47: change the context since achromatopsia and some forms of CSNB are considered now as slowly progressive.
We have taken this important point in the revised version of the manuscript and rephrase as following: " Achromatopsia (ACHM) and congenital stationary night blindness (CSNB) are two slowly progressive retinal diseases that can present in the same way at birth"
Introduction, row 57-58: “This, with the advent of therapies for these retinal diseases, gives molecular testing in these diseases an even higher relevance”. Please change this statement to e.g. “The importance of genetic diagnosis in the advent of recently evolving therapies for inherited retinal degenerations is warranted”.
We thank the Reviewer for his suggestion which we have included in the revised version of the ms.
Introduction, row 58-60: please, improve grammar in this statement.
We have rephrased as following: However, the diagnosis yield of the sequencing of coding regions and intron-exon boundaries of known genes is limited to 50 to 80%.
Introduction, row 68-69: please change to e.g. “Since the protein is almost exclusively expressed in the retina, pathologic variants in RPGRIP1 cause a non-syndromic form of LCA”
We thank the Reviewer for his suggestion which we have included in the revised version of the ms.
Introduction, row 78-80: change this statement to e.g. “In our study we made an attempt to AON treatment of the lymphoblasts obtained from patients carrying the change. We were able to correct the abnormal splicing, which could have an important insight into splice-switching therapy of this frequent RPGRIP1 disease allele (13%)”.
We thank the Reviewer for his suggestion which we have included in the revised version of the ms.
Results, row 254: remove this statement “(52 RPGRIP1 disease alleles, article in preparation)”.
Done
Results, row 287: “could not be determined for lack of parental DNA” – please change to “could not be determined due to absence of parental DNA”.
Done
Results, row 301: “compared human fetal retina” – change to “compared to human fetal retina”.
Done
Discussion, row 337: remove “(article in preparation)”.
Done
Discussion, row 353-354: “be it in homozygosity or in compound heterozygosity” – please change to “in homozygous or heterozygous state”.
Done
Discussion, row 373: “preservation foveal photoreceptors” – change to “preservation of foveal photoreceptors”.
Done
Discussion, row 374-376: change to “Antisense oligonucleotide (AON)-mediated splice modulation, which is a powerful method to correct the consequences of pathogenic variants affecting pre-mRNA splicing, is another attractive method to treat RPGRIP1 deep intronic variants”.
Again, we thank the Reviewer for his suggestion which we have included to improve our ms.
Discussion, row 381: change to “Furthermore, antisense oligonucleotides…”
Done
Discussion, row 385-387: please change this statement “In this light, the most frequent RPGRIP1 variation c.1468-128G>T is a very attractive candidate for antisense oligonucleotide (AON)-mediated splice-switching therapeutic developments”.
We thank the Reviewer for his suggestion which we have included in the revised version of the ms.
Discussion, row 396-397: change to: “Two of them presented a severe cone-rod dystrophy with the clinical features characteristic for RPGRIP1-associated retinal degeneration”.
We thank the Reviewer for his suggestion which we have included in the revised version of the ms.
Discussion, row 399: change “exist” to “exists”.
Done
Table 1: improve the table: separate “Gender” from the “cDNA and protein changes”, should appear as a separate column. Write variants according to latest HGVS recommendations. “Extra ocular features” – the table does not have upper border. Consider to change “flat rod and cone ERG” to “undetectable ERG”.
Table 1 has been modified according to the Reviewer recommendations
Table 2: improve borders of the table.
Done
Figure 2: text row 502 change to “cryptic”
Done
Figure 2A – is not well readable
We have added some annotations on the Figure, which we hope will help to understand.